# Financial Technology and Its Impact on Digital Literacy in India: Using Poverty as a Moderating Variable

Rahul Singh Gautam [1], Shailesh Rastogi [1] , Aashi Rawal [1], Venkata Mrudula Bhimavarapu [1,*] ,
Jagjeevan Kanoujiya [1] and Samaksh Rastogi [2]

1   Symbiosis Institute of Business Management, Symbiosis International (Deemed University), Pune 412115,
    India; bouddhrahul@gmail.com (R.S.G.); krishnasgdas@gmail.com (S.R.); aashi.rawal07@gmail.com (A.R.);
    jagjeevan24288@yahoo.co.in (J.K.)
2   Chartered Accountant & Certified Public Accountant (Australia), Managing Partner, Rastogi and Co
    Chartered Accountants, Navi Mumbai 400703, India; samaksh@finlexsolutions.com
*   Correspondence: mrudulabhimavarapu@gmail.com

**Abstract:** Financial technology is a powerful tool in financial infrastructure, used to strengthen and smooth the delivery of financial services into the broader space. Financial technology involves software, applications, and other technologies designed to improve and automate traditional forms of financial services for businesses established in different areas. The authors aimed to explore the impact of financial technology on the digital literacy rate in India, by utilizing the poverty score as a moderating variable. The panel data analysis (PDA) has been employed in the current study. Data from 29 states and two union territories (UTs) of India were considered for three financial years, i.e., 2017–2018 to 2019–2020. The study's findings reveal that Kisan Credit Cards (KCCs), both in terms of numbers and amount, are positively associated with the literacy rate. However, ATMs are negatively significant in association with literacy rate. Furthermore, the study's empirical results show that KCCs and ATMs positively impact literacy when interacting with poverty scores. The study's findings bring noteworthy implications for the government and other officials to understand the situation at the ground level of Indian states and UTs while forming new rules and policies for society's betterment, particularly in finance and digital literacy. Additionally, the findings imply that ordinary people living in urban and rural areas of India should take advantage of financial technology and get motivated towards digital literacy.

**Keywords:** financial technology; digital literacy; Kisan credit cards; ATMs

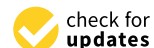



## 1. Introduction

The need to transform the traditional way of working in the financial services industry is increasing daily in response to the changing needs of people, who require easy and more efficient forms of technology in every area.

This brings the financial technology industry into the picture with technology-enabled financial solutions. FinTech, which stands for financial technology, is one of the most significant breakthroughs in the financial services sector and is driven by information technology, economic sharing, regulation, and policy (Lee and Shin 2018). FinTech is a financial industry composed of companies that deliver technology-based financial service systems for efficient and accessible financial services (McAuley 2015). The term FinTech is not confined to specific sectors (e.g., financing) or business models (e.g., peer-to-peer (P2P) lending), but instead covers the entire scope of services and products traditionally provided by the financial services industry (Arner et al. 2015).

FinTech has brought a revolution to the financial services industry. After the 2008 global financial crisis, improvements in e-finance and mobile technologies for financial organizations sparked the development of FinTech (Suryono et al. 2020). FinTech has excellent potential for financial services' business revenues, regulators' income, and the welfare

of customers (Anagnostopoulos 2018). Customers nowadays only crave efficient working devices that can help to save their time and money. With the help of financial technologies, companies can develop a customized form of software and applications to provide a better experience for their customers. However, even after developing innovative technologies, the hard work will be in vain if most of a country's population is unaware of how these technologies can be used. This makes literacy a significant factor in understanding the full and proper implementation and usage of FinTech.

The value of being literate in this world is not hidden from anyone. Literacy can be a basic knowledge of reading and writing, or even having the highest level of knowledge. However, nowadays, in the era of technological advancements, such basic knowledge of reading and writing cannot be fruitful until it includes technology and cognizance of other fields. Many authors have called attention to new or multiple literacies (Bazalgette 1988; Buckingham 1993; Tyner 1998; and many others). It is necessary to have knowledge in business, finance, and many other things to live a better and easier life, and having digital literacy is also one of them as being digitally literate is counted as one of the essential qualities that a person must attain in his or her life. Digital literacy is the ability to succeed in encounters with the electronic infrastructures and tools that make possible the twenty-first century world (Martin 2006). Digital financial literacy is critical at the present time. We know that now all financial services and products are available in digital form in almost all economies globally, including India; the present government is also focusing on cashless India and digital India (Prasad et al. 2018).

Nowadays, governments are creating initiatives to increase digital literacy among the citizens of their respective countries. As far as the Indian government is concerned, one of the latest campaigns is the Pradhan Mantri Gramin Digital Saksharta Abhiyan Yojana. Under this, the Government of India aims to provide digital literacy to 60 million citizens in rural India (Nedungadi et al. 2018). The Indian government is also promoting Digital India, which aims to ensure that government services are made available to citizens electronically by reducing paperwork (Goswami 2016). They have recently launched many schemes like Pradhan Mantri Jan Dhan Yojna, Jeevan Jyoti Bima Yojna, Suraksha Bima MUDRA Bank Yojna, and BHIM. The Vittiya Saksharta Abhiyan (VISAKA) has also been launched by the Ministry of Human Resources (Prasad et al. 2018).

The main reason behind such a heavy promotion of digital literacy is that everything in this world is becoming digitalized. This era of digitalization demands a person to be fully equipped with the knowledge of using digital equipment, machines, and other electronic devices in every city or country. The perfect example of such a typical digital machine is the ATM. These have been launched to ease the process of going into a bank to withdraw or deposit cash. The government has also installed these machines in urban and rural areas of the country. However, if people are unaware of using ATMs, all these expenditures are a waste of money and time; this is why various government departments and NGOs are undertaking digitalization awareness campaigns and other initiatives so that people can become aware of the facilities.

Awareness campaigns have a significant impact, mainly on the people living in rural areas. As most individuals in these areas are illiterate, they are unable to learn about the numerous government facilities that have been introduced. Rural and tribal people are among the most disadvantaged segments of society and are often exploited due to their illiteracy and lack of awareness (Nedungadi et al. 2018). Therefore, these campaigns may aid in raising their awareness and encouraging them to utilize such facilities. A country's government undertakes all such initiatives to improve the citizens' standard of living. Betterment is the only motive in the minds of the officials and governmental bodies.

No similar study that focused on discovering a direct association between FinTech and digital literacy has come to our attention, because the earlier literature on digital literacy has not focused much on the Indian economy. As we all know, the Indian economy is one of the fastest-growing economies worldwide; therefore, it is now becoming necessary to study each factor and area that can affect the proper and efficient growth of the Indian

economy. This research gap encouraged us to pursue the current study by including these two significant factors (FinTech and digital literacy).

There are two motivating factors that encourage pursuing a study in financial and digital literacy. Firstly, many researchers have conducted studies involving the concept of FinTech or digital literacy. However, as per our knowledge, none of them observed the impact of FinTech by considering essential elements such as ATMs and KCCs on the digital literacy of ordinary people, even though essential factors like these generally have a direct and significant impact on the day-to-day lives of common people. Secondly, no such study has been conducted that has used poverty as a moderating variable to find its influence on the association among ATMs, number of KCCs, amount of KCCs, and the literacy rate.

With this study, we have aimed to investigate the impact of financial technology on the level of digital literacy among the people living in India, as the Indian economy is one of the fastest-growing economies worldwide, and we have the second-largest population in the world. These reasons make it necessary to study each factor that can impact the efficient growth of the Indian economy. Two common or essential elements, such as ATMs and Kisan credit cards that can directly impact the everyday life of people, are considered independent variables, and the poverty score is a moderating variable for the current study. The study is conducted for three financial years, running from 2017–18 until 2019–20, and the objective we set for this study was to identify the impact of financial technology on digital literacy in 29 states and two union territories of India.

We believe this study delivers substantial implications to the concerned stakeholders in the following ways. Firstly, it helps the government and other NGOs to realize the need to start different types of campaigns to impart digital literacy to improve people's financial literacy, especially those living in rural areas all over the country. Secondly, the study contributes significantly to the literature on financial technology, as many studies have involved digital literacy. However, FinTech is a comparatively new topic, and more literature is needed in this area. After looking through this study, the interest of future researchers can be developed in the field of FinTech and financial inclusion. Finally, the study can give some basic ideas to the country's common people about the efforts of governmental bodies and others involved in providing innovative technologies in every part of the country.

The current study is structured in the following manner. Section 2 comprises a review of the literature and hypotheses formulation. Section 3 concentrates on research objectives, leading to data collection and methodology, followed by regression analysis with results in Section 4. A wide discussion is in Section 5, while conclusions and future scope conclude the paper in Section 6.

## 2. Literature Review and Hypotheses Formulation

### 2.1. FinTech and Financial Inclusion

The introduction of financial technology has significantly upgraded the financial services sector. In order to achieve easy implementation of financial inclusion programs at all levels, the government needs to adopt FinTech like biometric ATMs, mobile money, and others, and increase its usage by providing all the facilities in every district and village of the country. FinTech influences financial literacy, and financial literacy influences financial inclusion, affecting rural economic development (Rastogi et al. 2021.) In India, those aiming to achieve financial inclusion through FinTech can be useful if its use is not treated as a prerequisite (Gupta and Singh 2013). The old cash payment system in agriculture can be replaced by leveraging FinTech, digital business models, and the rapid adoption of mobile phones, as the provision of these facilities can help to ease the lives of the rural household (Babcock 2015).

FinTech has become a form of government assistance that provides financial services to the public at an affordable cost, leading to economic development (Rastogi et al. 2017; Sharma et al. 2020; Kuknor and Rastogi 2021). Financial services should be provided to the country in general for better economic development, and financial inclusion is significant

(Dixit and Ghosh 2013). Financial inclusion is an activity to gain access to financial services by eliminating various barriers (Lumenta and Worang 2019); it enables universal access to and coverage of financial services, so their benefits for the economy and development can be widely felt in various countries (Senyo et al. 2020). Financial inclusion ranks high among three states in India, and ninety percent of states have little/weak access to financial services. There is a pressing need to increase financial inclusion for economic development (Anand and Chhikara 2013).

Over the past decade, the Indian government has taken a big step towards financial inclusion, and recently, Brazil has made an infusion for India because they can provide a global primary income base, which will help to significantly reduce complete poverty in India (Maripally and Bridwell 2017). Financial inclusion is essential for economic and social development in the country, because it helps reduce poverty and wealth; the government constantly encourages banks and other financial institutions to increase their presence and activities in rural areas (Kumar and Gupta 2019).

### 2.2. FinTech and Digital Literacy

FinTech development may harm financial well-being by increasing the financial risk and fostering impulsive consumer behavior (Prasad et al. 2017; Shen et al. 2018; Panos and Wilson 2020). The gaps in the form of not having full critical literacy while using the FinTech products among rural and urban residents and income groups are extensive (Potrich et al. 2015; Prasad et al. 2017; Morgan et al. 2019). Hence, enhancing digital and financial literacy is essential to effectively utilize FinTech products and services.

There are many ways people can learn about software, applications, and other essential technologies. Apart from homes, NGO workers and other people employed by the government can go to different places such as schools, temples, parks, and many other familiar places to impart practical information about the new and innovative technologies they can use daily. The provision of digital literacy to improve life skills for people existing in civil society, schools, and various other institutions like government organizations can also bring significant change in society (Nedungadi et al. 2018; Bank 2020). Digital literacy is a crucial criterion for the usability of digital products and should be addressed urgently (Kollinal et al. 2019).

Digital Financial Literacy has become critical in this digital era (Gozgor et al. 2018), and in order to make wise financial decisions and eventually achieve individual financial wellbeing, one must possess the financial awareness, knowledge, skills, attitude, and behaviors required (OECD 2018). Digital technologies have facilitated the globalization of business operations and commercial cultural output, and additionally, they have made it feasible to accumulate vast amounts of personal data on people, who are now commodities (Martin and Grudziecki 2006). The development of digital technology has not only made it possible to communicate with individuals anywhere on the globe, but it has also made it possible to gather data on every single thing that exists on this planet.

Furthermore, various studies discuss financial technology or digital literacy; however, none explore the FinTech industry by dividing it into three standard variables (ATM, number of KCCs, and amount of KCCs). This situation implies a research gap in the FinTech area. Thus, the following alternate hypotheses are framed to empirically test the role of three factors on the literacy rate:

**Hypothesis H1.** *The number of ATMs positively impacts the literacy rate.*

**Hypothesis H2.** *The number of Kisan credit cards (KCCs) positively impacts the literacy rate.*

**Hypothesis H3.** *The amount of Kisan credit cards (KCCs) positively impacts the literacy rate.*

*2.3. FinTech, Digital Literacy, and Poverty*

The growth of FinTech has left many disadvantaged groups in the country. Overall development of a country requires an even distribution of all the technologies in urban and rural areas. The people living in rural areas know very little about FinTech and its services, and this gap in knowledge decreases the chances of poverty reduction. As poverty continues, rural people will remain less interested in learning new digital technologies, as they are not getting above their basic needs and they will never become interested in knowing about ATMs or the provision of Kisan credit cards by the government. Consequently, proper implementation and efficient planning are required to make people aware of the facilities launched for their betterment.

Digital literacy plays an essential role in economic development and poverty reduction, by helping people in rural areas to get the basic knowledge about various schemes started by the government, increasing their income, and leading to the overall development of rural areas. With the digitization of services and digital literacy provision through different campaigns, rural people can now access financial services (Tripathi and Dungarwal 2020). Furthermore, digitization of the economy needs to be done carefully as many illiterate rural people cannot access the internet, ATMs, or mobile banking services (Singh and Naik 2018. Efforts should be made to bring technology to the bottom of the pyramid to make India a digital economy.)

Digital literacy is a crucial criterion for the usability of digital products and should be addressed urgently (Kollinal et al. 2019). Digital India was launched by the Government of India to provide government schemes digitally and aims to add good internet connectivity and improve digital literacy. This will help economic growth and develop the country (Sushma and Kumar 2019). Encouraging digital literacy in disadvantaged groups through concerted initiatives is also necessary to achieve inclusive financial and economic growth (Liew et al. 2020).

The existing literature studies show that no research has attempted to connect financial technology with digital literacy in a vastly spread economy like India. This prompted the present authors to consider the gaps identified in the literature and motivated them to conduct the present study. In addition, poverty has been included as a moderating variable to make the results more conceptually stronger and creative. Poverty has an interactive effect on the relationship between the independent and dependent variables in the current investigation (FinTech and digital literacy, respectively). In order to fill this significant gap, the following hypotheses were formulated, to empirically test the impact of financial technology and digital literacy level in Indian states and UTs under the poverty reduction factor.

**Hypothesis H4a.** *The number of ATMs positively impacts the literacy rate under the influence of poverty score.*

**Hypothesis H5a.** *The number of Kisan credit cards positively impacts the literacy rate under the influence of poverty score.*

**Hypothesis H6a.** *The amount of Kisan credit cards positively impacts the literacy rate under the influence of poverty score.*

**3. Data and Methodology**

*3.1. Data Sourcing for the Study*

The study spans India's 29 states and two union territories (the Andaman and Nicobar Islands, and Puducherry), over three fiscal years (2017–2018 to 2019–2020), fixed at 93 observations. The only reason to consider 29 states and two UTs for the sample period is the availability of reliable data for consistent analysis of the new evidence of financial inclusion, with the technology base increasing day by day in India's financial system. The information was gathered from the RBI's (Reserve Bank of India) official websites and

other websites such as timesnownews.com. Table 1 presents a detailed description of the variables investigated for the study and the data sources.

**Table 1.** Variable Definition.

| Variable Name | Symbol | Description | Data Source |
|---|---|---|---|
| Literacy rate | lr | The literacy rate measures the percentage of people who are digitally literate about digital financial services. | www.timesnownews.com. Accessed on 16 October 2021 |
| Number of ATMs | l_atm | The natural logarithm of the number of ATMs considered for the study is situated in 29 states and two union territories in India. | RBI (2019) report |
| Number of Kisan credit cards | l_KCC_n | Natural logarithm number of Kisan Credit Cards (KCCs) from 29 Indian states and two union territories. | RBI (2020) report |
| Amount of Kisan credit cards | l_KCC_amt | Natural logarithm of the amount of Kisan Credit Cards (KCCs) from 29 states and two union territories of India | RBI (2020) report |
| The interaction term of poverty score and number of ATMs | l_ps_atm | The interaction term is obtained by multiplying the natural logarithm of poverty score and ATMs. (l_ps*l_atm) | – |
| The interaction term of poverty score and the number of Kisan credit cards | l_ps_KCC_n | The interaction term is obtained by multiplying natural logarithm of poverty score and number of Kisan credit cards (l_ps*l_KCC_n) | – |
| The interaction term of poverty score and amount of Kisan credit cards | l_ps_KCC_amt | The interaction term is obtained by multiplying natural logarithm of poverty score and amount of Kisan credit cards (l_ps*l_KCC_amt) | – |

Source: Authors Compilation. Note: Table 1 describes the variables considered for the present study.

### 3.2. Methodology Employed for the Study

The hypotheses testing is done using the panel data model in this study. Panel data analysis was chosen because it outperforms classical cross-sectional or time-series data analysis (Baltagi 2008; Wooldridge 2003; Hsiao 2007), and as a result, it has the potential to provide more information than a time series or a cross-section analysis. The panel data model is also a better choice due to its consistent estimates having less bias. The endogeneity issue does not exist, and so a static model is preferred for the analysis (Baltagi 2008; Wooldridge 2003). STATA 16 is used to run the study's static panel regression model. The following are the econometric model specifications (Wooldridge 2003; Baltagi 2008) used in the study:

$$\text{lr}_{it} = \alpha + \beta_1 \text{ l\_atm}_{it} + \text{u}_{it} \qquad \text{(Model 1)} \tag{1}$$

$$\text{lr}_{it} = \alpha + \beta_1 \text{ l\_kcc\_n}_{it} + \text{u}_{it} \qquad \text{(Model 2)} \tag{2}$$

$$\text{lr}_{it} = \alpha + \beta_1 \text{ l\_kcc\_amt}_{it} + \text{u}_{it} \qquad \text{(Model 3)} \tag{3}$$

$$\text{lr}_{it} = \alpha + \beta_1 \text{ l\_ps\_atm}_{it} + \text{u}_{it} \qquad \text{(Model 4)} \tag{4}$$

$$\text{lr}_{it} = \alpha + \beta_1 \text{ l\_ps\_kcc\_n}_{it} + \text{u}_{it} \qquad \text{(Model 5)} \tag{5}$$

$$\text{lr}_{it} = \alpha + \beta_1 \text{ l\_ps\_kcc\_amt}_{it} + \text{u}_{it} \qquad \text{(Model 6)} \tag{6}$$

where *lr* (literacy rate) is the dependent variable for all the models and *l_atm*, *l_KCC_n*, and *l_KCC_amt* (base variables in base models), i.e., Equations (1)–(3) *are explanatory variables*. *l_ps_atm*, *l_ps_KCC_n* and *l_ps_KCC_amt* (interaction-terms as discussed in Table 1) are the explanatory variables in interaction models i.e., Equations (4)–(6). Moreover, $uit = \mu i + vit$ ($\mu i$ denotes the individual effect that is unobservable, and $v_{it}$ specifies the remaining disturbance). $\beta$ is coefficient, and $\alpha$ is constant-term. 'it' shows state 'i' and time 't'.

### 3.3. Descriptive Statistics and Correlation Matrix

Tables 2 and 3 present the descriptive statistics and correlation of the variables considered for the study. In Table 2, the mean value of *lr* is 78.016, slightly close to Min, showing a moderate literacy level on average; *l_kcc_n* and *l_kcc_amt* have an average of 13.049 and 12.851, respectively (values are closer to Max). This indicates a substantial use of KCCs. From Table 2, it is also evident that the variables' standard deviation is in single-digit numbers, showing that the variables do not deviate much from the mean values of the variables considered for the study. This implies that the status of literacy rate, ATMs, KCC number, and amount are not varying much across sample states and union territories. From Table 3, it is perceived that all the six exogenous variables are positively significant to the dependent variable *lr*, denoting that the variables move in the same direction; this implies that if one variable increases, the other variable shows an increase in its value. All the variables show a *p*-value of (0.0000), as the significant values are more than 0.8, implying a multicollinearity issue (Baltagi 2008) exists. Therefore, each variable is tested individually using six different econometric models to overcome the issue of highly correlated exogenous variables, as suggested by (Wooldridge 2003; Baltagi 2008).

**Table 2.** Descriptive Statistics.

| | Descriptive Statistics | | | |
| --- | --- | --- | --- | --- |
| | **Mean** | **SD** | **Min** | **Max** |
| lr | 78.016 | 7.9937 | 61.8 | 96.2 |
| l_atm | 9.3425 | 1.6526 | 6.1003 | 11.612 |
| l_KCC_n | 13.049 | 2.4474 | 8.2941 | 16.301 |
| l_KCC_amt | 12.851 | 2.6376 | 7.3524 | 16.241 |
| l_ps_atm | 13.396 | 1.6338 | 10.165 | 15.891 |
| l_ps_KCC_n | 17.103 | 2.4199 | 12.206 | 20.275 |
| l_ps_KCC_amt | 16.905 | 2.6141 | 11.444 | 20.335 |

Source: Authors Compilation. Note: Table 2 explains the mean, standard deviation, and minimum and maximum values of the variables considered for the study.

**Table 3.** Correlation Matrix.

| | Correlation Matrix | | | | | | |
| --- | --- | --- | --- | --- | --- | --- | --- |
| | **lr** | **l_atm** | **l_KCC_n** | **l_KCC_amt** | **l_ps_atm** | **l_ps_KCC_n** | **l_ps_KCC_amt** |
| lr | 1 | | | | | | |
| l_atm | 0.9346 * (0.0000) | 1 | | | | | |
| l_KCC_n | 0.9356 * (0.0000) | 0.9165 * (0.0000) | 1 | | | | |
| l_KCC_amt | 0.9258 * (0.0000) | 0.9471 * (0.0000) | 0.9717 * (0.0000) | 1 | | | |
| l_ps_atm | 0.9120 * (0.0000) | 0.9927 * (0.0000) | 0.9054 * (0.0000) | 0.9391 * (0.0000) | 1 | | |
| l_ps_KCC_n | 0.9237 * (0.0000) | 0.9142 * (0.0000) | 0.9967 * (0.0000) | 0.9700 * (0.0000) | 0.9128 * (0.0000) | 1 | |
| l_ps_KCC_amt | 0.9133 * (0.0000) | 0.9439 * (0.0000) | 0.9670 * (0.0000) | 0.9972 * (0.0000) | 0.9449 * (0.0000) | 0.9713 * (0.0000) | 1 |

Source: Authors compilation. Note: Values in the correlation matrix are correlation coefficients. Values in parenthesis are *p*-values. * Significant at 5%.

## 4. Regression Analysis

### 4.1. Static Panel Data Regression Analysis

The panel data regression analysis is performed on all six models, and the results are reported in Tables 4–6, respectively. From Table 4, it is evident that models 1 and 2 have significant *p*-value for the F-test for fixed effects and the Breush–Pagan test for random

effects. Hence, the Hausman test determines the model's good fit. The Hausman test results suggest the random effect model is a good fit for model 1 as the *p*-value is more than 0.05. Whereas for model 2, as the *p*-value of the Hausman test is less than the value of 0.05, the fixed effect is chosen as the best fit for the model. The exogenous variables in models 1 and 2 are significant to the dependent variable (DV) *lr*, where *l_atm* is negatively significant (with coefficient −1.484 and *p*-value 0.04), implying an inverse relationship with literacy rate. Hence, it implies that an increment in ATMs lowers the literacy rate. In contrast, in model 2, *l_kcc_n* shows a positive association with the DV (with a coefficient of 0.560 with a *p*-value of 0.001), implying that Kisan credit cards increase the literacy rate among the farmers.

**Table 4.** Result of Regression Analysis (Static Panel Data Analysis).

| LR. | Model 1 | | | Model 2 | | |
|---|---|---|---|---|---|---|
| | Coef. | Std. Err | *p*-Value | Coef. | Std. Err | *p*-Value |
| constant | 91.88 * | 6.886 | 0.000 | 70.70 * | 1.929 | 0.000 |
| l_atm | −1.484 * | 0.7396 | 0.045 | – | – | – |
| l_kcc_n | – | – | – | 0.5608 * | 0.1478 | 0.001 |
| R-Square | | 0.1505 | | | 0.2396 | |
| SE of Regression | | 3.355 | | | 3.367 | |
| Note: No of observations (n) | | 93 | | | 93 | |
| Degree of freedom | | 61 | | | 61 | |
| F-test Fixed Effect | | 13.11 * (0.0000) | | | 11.88 * (0.0000) | |
| Random Effect Test | | 58.09 * (0.0000) | | | 52.31 * (0.0000) | |
| Hausman Test | | 2.36 (0.1242) | | | 5.73 (0.0167) | |
| Wald test for Heteroscedasticity [1] | | 24,678.60 * (0.0000) | | | $4.4 \times 10^5$ * (0.0000) | |
| Wooldridge Autocorrelation Test [2] AR (1) | | 3.554 (0.0691) | | | 9.713 * (0.0040) | |

**Note**: [1] Wald test of heteroscedasticity has the null of no heteroscedasticity. [2] Wooldridge test of autocorrelation in the panel has the null of no autocorrelation (with one lag). SE of regression denotes the standard error. Theta estimates the random effect model (higher is better).Standard Errors are robust estimates due to significant heteroscedasticity, autocorrelation, or both.* symbolizes the significant coefficient values

Table 5 shows that the two exogenous variables, i.e., *l_kcc_amt* and *l_ps_atm* from models 3 and 4, are positively significant, with the DV literacy rate having coefficients of 0.996 and 8.186, respectively, with *p*-values < 0.05. The results of models 3 and 4 show that the amount of Kisan credit cards enhances literacy. Furthermore, while there is a higher poverty rate, the increasing number of ATMs enhances literacy among the farmers, indicating that the high poverty rate and increase in the infrastructure of the financial services will lead to an improvement in digital literacy in rural India. It should be noted that model 3 adopts the random effects model; in contrast, model 4 has adopted fixed effects as per the results obtained from the Hausman test. The Hausman test was performed as both models show that the F-test for fixed effects and the Breush–Pagan test for random effects have a significant *p*-value.

Table 6 presents the regression results for models 5 and 6. For both models, the Hausman test reveals that the Fixed effect is the best fit for models 5 and 6 (as in having significant *p*-values < 0.05). Both models' F-test for fixed effects and the Breush–Pagan test for random effects have a significant *p*-value; thus, the Hausman test is performed. The interaction terms of poverty score with the number and amount of Kisan credit cards show a significant positive association (as the coefficient of the interacting variable has a positive and significant *p*-value at 5% significance) with the dependent variable literacy rate (having coefficients of 1.198 and 2.709, respectively, with *p*-values < 0.05), implying that the improvement of the literacy rate among the farmers is achieved by the increase in the usage of Kisan credit cards under higher poverty scores. The detailed explanation of the interaction term is explained in the subsequent Section 4.2.

**Table 5.** Result of Regression Analysis (Static Panel Data Analysis).

| LR. | Model 3 | | | Model 4 | | |
|---|---|---|---|---|---|---|
| | Coef. | Std. Err | *p*-Value | Coef. | Std. Err | *p*-Value |
| Constant | 65.21 * | 4.376 | 0.000 | −31.64 | 45.10 | 0.488 |
| l_kcc_amt | 0.9962 * | 0.3405 | 0.006 | – | – | – |
| l_ps_atm | – | – | – | 8.186 * | 3.367 | 0.021 |
| R-Square | | 0.2008 | | | 0.1164 | |
| SE of Regression | | 3.367 | | | 2.920 | |
| Note: No of observations (n) | | 93 | | | 93 | |
| Degree of freedom | | 61 | | | 61 | |
| F-test Fixed Effect | | 12.29 * (0.0000) | | | 18.88 * (0.0000) | |
| Random Effect Test | | 55.85 * (0.0000) | | | 57.20 * (0.0000) | |
| Hausman Test | | 3.02 (0.0823) | | | 19.67 * (0.0000) | |
| Wald test for Heteroscedasticity [1] | | $3.8 \times 10^5$ * (0.0000) | | | 56,860.14 * (0.0000) | |
| Wooldridge Autocorrelation Test [2] AR (1) | | 9.933 * (0.0037) | | | 2.466 (0.1268) | |

**Note**: [1] Wald test of heteroscedasticity has the null of no heteroscedasticity. [2] Wooldridge test of autocorrelation in the panel has the null of no autocorrelation (with one lag). SE of regression denotes the standard error. Theta estimates the random effect model (higher is better). Standard Errors are robust estimates due to significant heteroscedasticity, autocorrelation, or both. * symbolizes the significant coefficient values.

**Table 6.** Result of Regression Analysis (Static Panel Data Analysis).

| LR. | Model 5 | | | Model 6 | | |
|---|---|---|---|---|---|---|
| | Coef. | Std. Err | *p*-Value | Coef. | Std. Err | *p*-Value |
| Constant | 57.53 * | 0.5065 | 0.000 | 32.22 | 21.18 | 0.139 |
| l_ps_kcc_n | 1.198 * | 8.662 | 0.025 | – | – | – |
| l_ps_kcc_amt | – | – | – | 2.709 * | 1.252 | 0.039 |
| R-Square | | 0.2119 | | | 0.1764 | |
| SE of Regression | | 3.292 | | | 3.210 | |
| Note: No of observations (n) | | 93 | | | 93 | |
| Degree of freedom | | 61 | | | 61 | |
| F-test Fixed Effect | | 13.11 * (0.0000) | | | 14.27 * (0.0000) | |
| Random Effect Test | | 51.86 * (0.0000) | | | 55.18 * (0.0000) | |
| Hausman Test | | 9.41 * (0.0022) | | | 10.50 * (0.0012) | |
| Wald test for Heteroscedasticity [1] | | $2.7 \times 10^5$ * (0.0000) | | | $1.2 \times 10^5$ * (0.0000) | |
| Wooldridge Autocorrelation Test [2] AR (1) | | 3.000 (0.0935) | | | 2.068 (0.1607) | |

**Note**: [1] Wald test of heteroscedasticity has the null of no heteroscedasticity. [2] Wooldridge test of autocorrelation in the panel has the null of no autocorrelation (with one lag). SE of regression denotes the standard error. Theta estimates the random effect model (higher is better). Standard Errors are robust estimates due to significant heteroscedasticity, autocorrelation, or both. * symbolizes the significant coefficient values.

### 4.2. Interaction Graphs

An interaction graph helps to show the impact of moderating variables on the association between the variables. In Figures 1–3, the solid lines indicate the high poverty score (***ps***), and long dotted lines indicate the low poverty score (***ps***). For all the interaction terms, poverty score ***ps*** acts as the moderator, and ***l_atm*** (Figure 1), ***kcc_n*** (Figure 2), and ***kcc_amt*** (Figure 3) are the moderated variables, respectively.

From Figure 1, it is evident that when the number of ATMs (***l_atm***) increases and the poverty score is high, the fall in the literacy rate is less (as the solid line is less steep) compared to the low poverty score. Furthermore, Figures 2 and 3 also indicate that when the number of Kisan credit cards (***kcc_n***) and the amount of Kisan credit cards (***kcc_amt***) escalates and the poverty score is high, the plunge in the literacy rate is slighter or less.

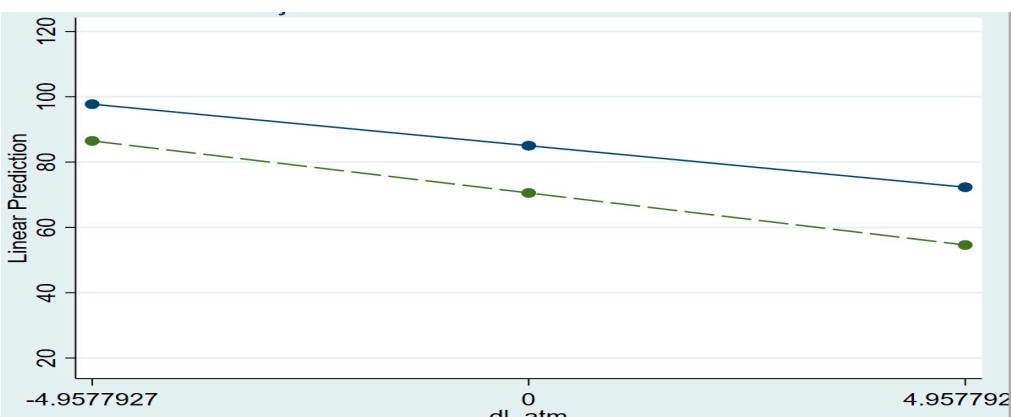

**Figure 1. Interaction graph of l_ps_atm.** Source: graph created using STATA 16. Note: The solid line of the graph represents the moderating variables' hig−level impact, whereas the long dash line represents the low impact level.

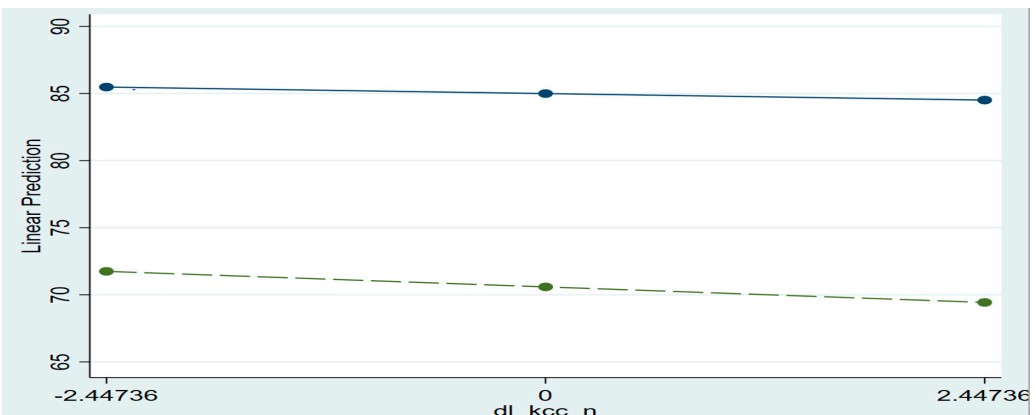

**Figure 2. Interaction graph of l_ps_kcc_n.** Source: graph created using STATA 16. Note: The solid line of the graph represents the moderating variables' high−level impact, whereas the long dash line represents the low impact level.

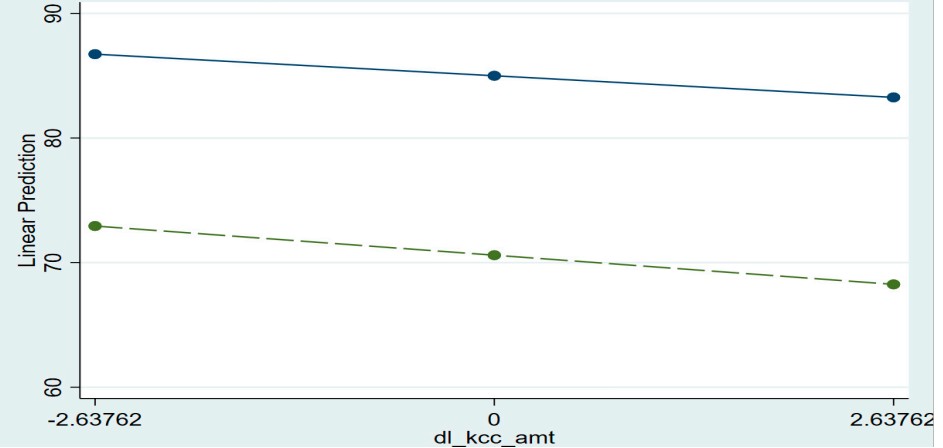

**Figure 3. Interaction graph of l_ps_kcc_amt.** Source: graph created using STATA 16. Note: The solid line of the graph represents the high impact of the moderating variables, whereas the long dash line represents the low impact level.

### 4.3. Robustness of the Regression ResultsRobustness Tests Determine How Specific Core

*Regression coefficients behave when the regression specification changes by adding or removing Regressors. If the estimated regression coefficients' signs and magnitudes are also plausible, it is commonly assumed that the estimated regression coefficients can be reliably interpreted as the true causal effects of the associated regressors (Lu and White 2014).*

The robustness of the current study's results is tested using STATA's CHECKROB command. It runs a series of regression models with varying core variable values; the results are shown in Table 7. It is evident from the PercSigni and Perc- have value '1' for the model amount of Kisan credit cards that it has a significant but negative relation coefficient with 100% share. Whereas PercSigni is '0' for the interaction term l ps atm and Perc+ is '1', implying that the association is positive but insignificant. The variables are insignificant in the remaining models 1,2, 5, and 6, and the association is negative because Perc- has a value of '1'.

**Table 7.** Robustness of the results.

| Core Variable | Max | Min | Mean | Avg STD | Perc Signi | Perc+ | Perc- | AvgT |
|---|---|---|---|---|---|---|---|---|
| l_atm | −1.4844 | −1.4844 | −1.4844 | 0.7632 | 0 | 0 | 1 | 1.9451 |
| l_kcc_n | −0.7905 | −0.7905 | −0.7905 | 0.4189 | 0 | 0 | 1 | 1.887 |
| l_kcc_amt | −0.9765 | −0.9765 | −0.9765 | 0.4471 | 1 | 0 | 1 | 2.1838 |
| l_ps_atm | 0.0762 | 0.0762 | 0.0762 | 0.8252 | 0 | 1 | 0 | 0.0923 |
| l_ps_kkc_n | −0.3921 | −0.3921 | −0.3921 | 0.4256 | 0 | 0 | 1 | 0.9215 |
| l_ps_kkc_amt | −0.5037 | −0.5037 | −0.5037 | 0.4664 | 0 | 0 | 1 | 1.0799 |

Source: Authors compilation. Note: Cor var stands for core variables. The terms Max, Min, and Mean refer to the coefficient's maximum, minimum, and mean values across all regressions. AvgSTD is an abbreviation for average standard deviation. PerSigni denotes the proportion of regression in which the coefficient is significant. Perc+ and Perc- represent the proportion of regression where the coefficient is positive or negative, respectively. AvgT denotes the average T-value.

### 4.4. Test Results for Endogeneity

The authors initiated the test for endogeneity for the explanatory variables in the model to confirm that these variables considered for the study are exogenous. All the explanatory variables *l_atm*; *l_KCC_n*; *l_KCC_amt*; *l_ps_atm*; *l_ps_KCC_n*; *l_ps_KCC_amt* have insignificant *p*-values (>0.05), both the Durbin Chi-square and Wu Hausman test, respectively, as demonstrated in Table 8. The *p*-values obtained from the tests confirm no endogeneity issue. Hence all the variables considered for the study are exogenous. Third lag value of the possible endogenous variable is taken as the instrument variable for performing the endogeneity test.

**Table 8.** Tests for Endogeneity.

| Test | Model 1 | Model 2 | Model 3 | Model 4 | Model 5 | Model 6 |
|---|---|---|---|---|---|---|
| Durbin Wu-Hausman chi2 test | 0.04792 (0.8267) | 0.084806 (0.7709) | 0.919438(0.3376) | 0.018272 (0.8925) | 0.122091 (0.7268) | 1.03308 (0.3094) |
| Durbin Wu-Hausman F- test | 0.043349 (0.8366) | 0.076809 (0.7837) | 0.855844 (0.3628) | 0.016514 (0.8987) | 0.110712 (0.7418) | 0.965276 (0.3343) |

Source: Author's Compilation. Note: Table 8 presents the test results of endogeneity for the variables studied, the values in the parenthesis represent the *p*-values of the tests.

## 5. Discussion

A negative relationship has been found between ATMs and the literacy level among people living in India. Based on this finding, it can be said that the first hypothesis $H_{1a}$ (positive association between ATMs and literacy rate) is rejected. The second hypothesis H2a formulated in this study, cannot be rejected; a positively significant relationship exists between the number of KCCs and the literacy rate. The third hypothesis H3a, that a positive association exists between the amount of KCCs and the literacy rate, cannot be rejected.

Looking for moderating effect of poverty score, the fourth hypothesis H4a (*The number of ATMs positively impacts the literacy rate under the influence of poverty score*) has enough evidence in its support. In the same way, the fifth H5a, and the sixth H6a hypotheses cannot be rejected, which means that a positive and significant association between the number and the amount of KCCs exists with the literacy rate where the poverty score is positively moderating the association. As a result, the poverty score can be considered a significant factor because it moderates the relationship between FinTech and digital literacy.

The current study has found different types of results regarding the factors which are being observed. There are not many studies conducted that include FinTech and digital literacy levels. If we discuss the first variable considered in the study, the association between ATMs and literacy rate depicts the association's negative nature. This might be because the usage of ATMs is not prevailing much among the people now, irrespective of the increasing number of ATMs. Secondly, a positive relationship of the number and the amount of KCCs exists to the literacy rate. Finally, there is a positive and substantial association between ATMs, the number of KCCS, the amount of KCCs, and the literacy rate where the moderator poverty score moderates the association. These connections are unique and are not seen in any other published work.

In comparison, many other studies have included digital literacy and FinTech variables for financial inclusion (Cahyani et al. 2021; Pangrazio 2016; Morgan et al. 2019, and many others). However, none of the studies used a moderating variable (poverty score). The current study's findings also show that financial institutions' technology-based products and services improve digital literacy among people. These findings are very exciting and novel in financial inclusion practices. Thus, the authors believe that the current study is novel in its approach by looking at the association of financial technology and literacy from different angles to deliver fresh and strong evidence. The current findings significantly augment the existing knowledge in financial inclusion, emphasizing the importance of technology, poverty score, and literacy.

The study has many important implications for the governmental bodies and common people of developing economies worldwide. Some of them are as follows: firstly, government and other officials can use this study to understand the ground-level situation and form some new policies they feel are needed in today's era to influence digital literacy level and financial technologies; secondly, after going through this study, ordinary people living in urban and rural areas of India should understand that these standard technologies like ATMs and cards significantly improve their digital literacy. The findings also have important implications for academics to keep technological aspects as an essential mode of providing education to adapt to technology smoothly. Thus, the current findings recommend that technological infrastructure has a key importance in improving literacy and should be given due consideration in such financial inclusion issues.

## 6. Conclusions and Future Scope

Technological innovation is spreading rapidly in every sector, particularly in the finance industry. Therefore, digital literacy knowledge becomes a critical issue in coping with such technological changes. This study has attempted to investigate the influence of financial technology (considering factors like the number of ATMs deployed, number of KCCs, and amount of KCCs) on the digital literacy of people living in Indian states and UTs. Moreover, this association is also investigated under the moderating effect of poverty. The final results show different impacts of each factor taken in the study. Regarding the relationship between ATMs and literacy rate, an inverse relationship has been found in the analysis. Secondly, a positive relationship exists between KCCs number and literacy rate and between the amount of KCCs and literacy rate. In addition, it is evident that under the influence of a poverty score, the relationship between all the three factors (ATMS, number of KCCs, and amount of KCCS) and literacy rate is positively significant. It implies that such technological facilities encourage the people to use them, which leads to improved literacy. Hence, the findings provide insights for all the government officials and departments

working in the digitalization area, NGOs, and many others for their decision-making on forming future policies and rules regarding the provision of financial literacy in different areas. The findings also imply that technical infrastructure is the key factor in improving literacy status. Furthermore, the current findings also have noteworthy implications for ordinary people to get engaged in learning technology skills to utilize better financial services. The current findings significantly contribute to the existing knowledge body in financial technology and literacy aspects by fresh corroboration.

This study finds interesting evidence on financial technology and digital literacy. However, the coverage of a small sample period is a limitation of the study. Hence, it opens doors for further investigations. On the availability of data, the number of ATMs and the number of KCC per capita should be taken as future scope of investigation for its impact on literacy. Other common factors directly or indirectly depict the presence or absence of financial literacy and digitalization around the globe. Researchers can focus on a particular state or city for a better and deep level of study, and can also increase the time to be covered in their future studies.

**Author Contributions:** Conceptualization, S.R. (Shailesh Rastogi) and R.S.G.; methodology, V.M.B.; software, V.M.B.; validation, S.R. (Samaksh Rastogi) and R.S.G. and V.M.B.; formal analysis, A.R.; investigation, J.K.; resources, V.M.B.; data curation, R.S.G.; writing—original draft preparation, S.R. (Shailesh Rastogi); writing—review and editing, V.M.B. and J.K.; visualization, S.R. (Samaksh Rastogi) and A.R.; supervision, S.R. (Shailesh Rastogi) and V.M.B.; project administration, A.R.; funding acquisition, S.R. (Shailesh Rastogi), S.R. (Samaksh Rastogi), R.S.G., V.M.B., J.K. and A.R. All authors have read and agreed to the published version of the manuscript.

**Funding:** This research received no external funding.

**Institutional Review Board Statement:** Not applicable.

**Informed Consent Statement:** Not applicable.

**Data Availability Statement:** Reserve Bank of India reports and www.timesnownews.com.

**Conflicts of Interest:** The authors declare no conflict of interest.

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
