# Peer review of "Financial Technology and Its Impact on Digital Literacy in India: Using Poverty as a Moderating Variable"

_jrfm, doi:10.3390/jrfm15070311_

Round 1

Reviewer 1 Report

This is a solid paper that studies literacy rate in India using the number of ATMs and the number/amounts in credit cards. 

I am concerned that financial literacy includes aspects such as familiarity with financial markets and portfolio analysis that are not explained by the number of ATMs and the number/amounts in credit cards. Can the authors expand on their methodology to see how this concern is mitigated?

Author Response

Reviewer 1 recommendation

English language and style are fine/minor spell check required 

Response from authors

Apologies for the inconvenience caused.  With professional Grammarly help we authors have checked and improvised accordingly.

Reviewer 1 recommendation

Does the introduction provide sufficient background and include all relevant references? - YES

Response from authors

Thanks for your kind and encouraging words.

Reviewer 1 recommendation

Are all the cited references relevant to the research? - YES

Response from authors

Thanks for your kind and encouraging words.

Reviewer 1 recommendation

Is the research design appropriate? – YES

Response from authors

Thanks for your kind and encouraging words.

Reviewer 1 recommendation

Are the methods adequately described? – YES

Response from authors

Thanks for your kind and encouraging words.

Reviewer 1 recommendation

Are the results clearly presented? – YES

Response from authors

Thanks for your kind and encouraging words.

Reviewer 1 recommendation

Are the conclusions supported by the results? – YES

Response from authors

Thanks for your kind and encouraging words.

Reviewer 1 recommendation

This is a solid paper that studies literacy rate in India using the number of ATMs and the number/amounts in credit cards.

Response from authors

Thanks for your kind and encouraging words.

Reviewer 1 recommendation

I am concerned that financial literacy includes aspects such as familiarity with financial markets and portfolio analysis that are not explained by the number of ATMs and the number/amounts in credit cards. Can the authors expand on their methodology to see how this concern is mitigated?

Response from authors

Thanks for pointing out the issue. Respected reviewer, to bring to your kind notice, the concept of the current research paper deals with is to explore the impact of financial technology on the digital literacy rate in India by utilizing the poverty score as a moderating variable. To explain digital literacy in specific means the knowledge of the rural people understandability and capability to use the digital aspects such as using kisan credit cards and ATMs has been considered as the criteria for the present research study.

Reviewer 2 Report

I find the findings are interesting. I have the following comments for the authors to improve their paper:

1) The authors should state the motives of their study and tell readers why their study is important and useful to academics and practitioners and state clearly their contributions to the literature in the introduction section.

2) The authors may discuss all important works and pioneer works and some of the recent studies related to their study

3) They may consider including a theory section to discuss the theory to support their model.

4) The authors should discuss the data more and discuss all variables being used.

5) The methodology section is too brief, the authors should discuss more on the methodology used in their paper. They should cite some papers for all the equations they are using in their paper.

6) The authors should discuss their findings and draw more inferences from their findings.

7) The conclusion section is too brief. They should discuss clearly the motives of their study, tell readers why their study is important and useful to academics and practitioners, state clearly their contributions to the literature in the conclusion section. They should point out the limitations of their approach and suggest directions for further study.

Author Response

Reviewer 2 recommendation

Does the introduction provide sufficient background and include all relevant references? -  Can be Improved

Response from authors

Thanks for the recommendation; considering the suggestions, the authors have updated the introduction section accordingly.

Reviewer 2 recommendation

Are all the cited references relevant to the research? -  Can be Improved 

Response from authors

Thanks for the valuable recommendation; considering the suggestions, the authors have updated the appropriate sections with recent articles along with the citations accordingly.

Reviewer 2 recommendation

Is the research design appropriate? -  Can be Improved     

Response from authors

Thank you for highlighting this issue. The research design’s corresponding sections are updated per the reviewer’s suggestion.

Reviewer 2 recommendation

Are the methods adequately described? -  Can be Improved          

Response from authors

Thanks for the recommendation; considering the suggestions, the authors have updated the methodology section accordingly.

Reviewer 2 recommendation

Are the results clearly presented? -  Can be Improved       

Response from authors

Thanks for the valuable suggestion made by the reviewers. We authors have updated the results section accordingly.

Reviewer 2 recommendation

Are the conclusions supported by the results? -  Can be Improved

Response from authors

Thanks for the valuable suggestion from the reviewers. We authors considering the recommendations now that the conclusion section has been updated.

Reviewer 2 recommendation

The authors should state the motives of their study and tell readers why their study is important and useful to academics and practitioners and state clearly their contributions to the literature in the introduction section.  

Response from authors

Thank you very much for highlighting this. Now the authors have explained in detail the motive for conducting the study. The novelty of the current study is manifold. To describe them in detail for the research conducted; First, the study adopted a multi-model approach to ensure strong evidence. Second, the investigation of financial technology and literacy under the moderating effect of poverty is unique as no such study is available in the literature. Hence, we believe that this study is novel in its approach and provide fresh and robust evidence. The same is updated in appropriate sections as per the reviewer’s suggestion.

Reviewer 2 recommendation

The authors may discuss all important works and pioneer works and some of the recent studies related to their study  

Response from authors

Thanks for the valuable recommendation; considering the suggestions, the authors have updated the appropriate sections with recent articles along with the citations accordingly.

Reviewer 2 recommendation

They may consider including a theory section to discuss the theory to support their model.

Response from authors

Thanks for the recommendation; considering the suggestions, the authors have updated the literature section accordingly.

Reviewer 2 recommendation

The authors should discuss the data more and discuss all variables being used.      

Response from authors

Thanks for the recommendation; considering the suggestions, the authors have updated the explanation of the data and methodology section accordingly.

Reviewer 2 recommendation

The methodology section is too brief, the authors should discuss more on the methodology used in their paper. They should cite some papers for all the equations they are using in their paper.

Response from authors

Thanks for the recommendation; considering the suggestions, the authors have updated the methodology section accordingly.

Reviewer 2 recommendation

The authors should discuss their findings and draw more inferences from their findings.

Response from authors

Thank you for pointing this out. We have now updated the discussion and implications sections accordingly.

Reviewer 2 recommendation

The conclusion section is too brief. They should discuss clearly the motives of their study, tell readers why their study is important and useful to academics and practitioners, state clearly their contributions to the literature in the conclusion section. They should point out the limitations of their approach and suggest directions for further study.

Response from authors

Thanks for the valuable suggestion from the reviewers. We authors considering the recommendations now that the conclusion section along with limitations and future scope for the study has been updated.

Reviewer 3 Report

From the overall presentation I would say that interesting research work has been done. The topic is also important for the readers of the journal. However, I have a few more significant challenges with the paper. 

  • The research methods used are appropriate but have limitations, and this should be mentioned. 

  • How “Literacy rate” is measured? Please include available Data Source (www.timesnownews.com is unavailable). 

  • You should use Number of ATMs per capita as explanatory variable.

  • The novelty of the paper is not sufficiently highlighted by the authors throughout the paper, and in the conclusions section. The novelty of the paper lies only in the geographical context of the research, where research is sparse and limited. 

  • The discussion and implications are rather short, and they should be extended. You need to improve the practical and academic implications. However, the paper has to underline the limits of the research. 

  • English language and style are fine/minor spell check required [For example, “Scheme 16. Note: Solid line of the grfaph represents” (page 9)].

Author Response

Reviewer 3 recommendation

English language and style are fine/minor spell check required       

Response from authors

Apologies for the inconvenience caused. Thank you for pointing this out. With professional Grammarly, we authors have checked and improvised accordingly.

Reviewer 3 recommendation

Does the introduction provide sufficient background and include all relevant references? -  Can be Improved

Response from authors

Thanks for the recommendation; considering the suggestions, the authors have updated the introduction section accordingly.

Reviewer 3 recommendation

Are all the cited references relevant to the research? -  Can be Improved

Response from authors

Thanks for the valuable recommendation; considering the suggestions, the authors have updated the appropriate sections with recent articles along with the citations accordingly.

Reviewer 3 recommendation

Is the research design appropriate? -  Must be Improved   

Response from authors

Thank you for highlighting this issue. The research design’s corresponding sections are updated per the reviewer’s suggestion.

Reviewer 3 recommendation

Are the methods adequately described? -  Can be Improved          

Response from authors

Thanks for the recommendation; considering the suggestions, the authors have updated the methodology section accordingly.

Reviewer 3 recommendation

Are the results clearly presented? -  Can be Improved       

Response from authors

Thanks for the valuable suggestion made by the reviewers. We authors have updated the results section accordingly.

Reviewer 3 recommendation

Are the conclusions supported by the results? -  Must be Improved

Response from authors

Thanks for the valuable suggestion from the reviewers. We authors considering the recommendations now that the conclusion section has been updated.

Reviewer 3 recommendation

The research methods used are appropriate but have limitations, and this should be mentioned.

How “Literacy rate” is measured? Please include available Data Source (www.timesnownews.com is unavailable).  

Response from authors

We are extremely sorry for this. It may be the case that the link is inactive.  The literacy rate for the study is measured as the percentage of people who are literate about digital financial services such as using the ATMs and Kisan credt cards.

Furthermore, from the link, as of now can only access the data for the financial year 2020.  We have now provided the Google Drive link for the used data. https://drive.google.com/drive/folders/19p-qLerkWrbKc0EwctD8BLICEpmAcA9f?usp=sharing

Reviewer 3 recommendation

You should use Number of ATMs per capita as explanatory variable.        

Response from authors

Thank you very much for suggesting this. Due to the unavailability of proper data for the sample period, we cannot consider it.  However, we will keep this in the future scope of the study.

Reviewer 3 recommendation

The novelty of the paper is not sufficiently highlighted by the authors throughout the paper, and in the conclusions section. The novelty of the paper lies only in the geographical context of the research, where research is sparse and limited.

Response from authors

Thank you very much for highlighting the point. The novelty of the current study is manifold. To describe them in detail for the research conducted; First, the study adopted a multi-model approach ensures strong evidence. Second, the investigation of financial technology and literacy under the moderating effect of poverty is unique as no such study is available in the literature. Hence, we believe that this study is novel in its approach and provide fresh and robust evidence. The same is updated in appropriate sections as per the reviewer’s suggestion.

Reviewer 3 recommendation

The discussion and implications are rather short, and they should be extended. You need to improve the practical and academic implications. However, the paper has to underline the limits of the research.  

Response from authors

Thank you for pointing this out. We have now updated the discussion and implications sections accordingly.

Reviewer 3 recommendation

English language and style are fine/minor spell check required [For example, “Scheme 16. Note: Solid line of the grfaph represents” (page 9)].      

Response from authors

Thank you for pointing this out. Extreme apologies for the inconvenience caused, with professional Grammarly we authors have checked and improvised accordingly.

Round 2

Reviewer 1 Report

The revised version is acceptable, congratulations!

Reviewer 2 Report

Thank you for revising the manuscript and for your responses.